# Effect of Food Restriction on Food Grinding in Brandt’s Voles

**DOI:** 10.3390/ani13213424

**Published:** 2023-11-05

**Authors:** Xin Dai, Yu-Xuan Han, Qiu-Yi Shen, Hao Tang, Li-Zhi Cheng, Feng-Ping Yang, Wan-Hong Wei, Sheng-Mei Yang

**Affiliations:** 1College of Bioscience and Biotechnology, Yangzhou University, 48 East Wenhui Road, Yangzhou 225009, China; daixin@yzu.edu.cn (X.D.); mx120211045@stu.yzu.edu.cn (Y.-X.H.); dx120230216@stu.yzu.edu.cn (Q.-Y.S.); mx120221091@stu.yzu.edu.cn (H.T.); mz120211667@stu.yzu.edu.cn (L.-Z.C.); fpyang@yzu.edu.cn (F.-P.Y.); whwei@yzu.edu.cn (W.-H.W.); 2Jiangsu Co-Innovation Center for Prevention and Control of Important Animal Infectious Diseases and Zoonoses, Yangzhou University, Yangzhou 225009, China

**Keywords:** food grinding, Brandt’s voles, food restriction, short-chain fatty acids, fecal microbiota

## Abstract

**Simple Summary:**

Food grinding is a behavior that occurs in rodents, supposedly affected by a variety of circumstances. It is yet unclear how these elements affect food grinding. The purpose of this study was to investigate the relationship between the gut microbiota and food grinding, as well as the impact of varying food supplies on this behavior. Our research showed that food restriction may change the abundance of the gut microbiota and its metabolites as well as reduce the amount of ground food to a larger extent than its effect on food consumption. Therefore, the inhibition of food grinding in Brandt’s voles may be attributed in part to differences in the abundance of gut microbiome and their metabolite, short-chain fatty acids, which is induced by food restriction. This study suggests that reducing the food supply could effectively inhibit food grinding during laboratory rodent feeding.

**Abstract:**

Food grinding is supposed to be influenced by multiple factors. However, how those factors affecting this behavior remain unclear. In this study, we investigated the effect of food restriction on food grinding in Brandt’s voles (*Lasiopodomys brandtii*), as well as the potential role of the gut microbiota in this process, through a comparison of the variations between voles with different food supplies. Food restriction reduced the relative amount of ground food to a greater extent than it lowered the relative food consumption, and altered the abundance of *Staphylococcus*, *Aerococcus*, *Jeotgalicoccus*, and *Un--s-Clostridiaceae bacterium GM1*. Fecal acetate content for the 7.5 g-food supply group was lower than that for the 15 g-food supply group. Our study indicated that food restriction could effectively inhibit food grinding. Further, *Un--s-Clostridiaceae bacterium GM1* abundance, *Aerococcus* abundance, and acetate content were strongly related to food grinding. Variations in gut microbial abundance and short-chain fatty acid content induced by food restriction likely promote the inhibition of food grinding. These results could potentially provide guidance for reducing food waste during laboratory rodent maintenance.

## 1. Introduction

Food grinding, also referred to as food spilling or food wasting, is a behavior of captive rodents that involves grinding food into orts that are left uneaten on the floor of cages [1,2], or a behavior of wild herbivorous rodents involving cutting and discarding substantial amounts of grass or herbaceous plant material [3,4]. Food grinding is regarded as a characteristic or obsessive behavior that occurs owing to the tedium of the environment or one that is stimulated by an optimal food-intake strategy, in which rodents choose a portion of food components through grinding to render their net energy intake as large as possible [1,5]. Food grinding is also affected by the hardness of food [1,6], genetic components [4], sleep deprivation [7,8], diet quality (cellulose content), and caloric restriction (food supply) [1]. However, the factors and mechanisms affecting food grinding are still poorly understood. Food scarcity can induce alterations in rodent behavior, such as decreased activity, food hoarding, and an increase in food foraging [9]. Specifically, how the food supply influences food grinding and its underlying mechanisms are not clear. Understanding the influencing factors and mechanisms of food grinding is important because this behavior results in a large amount of forage being wasted during laboratory rodent feeding. Ground food, as the leftovers of wild rodents, also plays an important role in grassland ecosystem functioning [3].

Brandt’s voles (*Lasiopodomys brandtii*), a small herbivorous mammalian species, inhabit the grasslands of Inner Mongolia, China [10,11,12,13]. This species consumes a diverse array of herbaceous plants, displaying a particular preference for *Leymus chinensis*, *Medicago varia*, *Stipa krylovii*, and *Saussurea runcinata* [14,15]. Notably, wild Brandt’s voles engage in wasteful behavior, gnawing on grasses without actually consuming them in the field. Remarkably, the quantity of grass wasted through non-feeding activity (excluding nesting material) accounts for a quarter of their daily food intake [16]. These feeding patterns have been shown to adversely impact the grassland ecosystems and livestock husbandry within the region [10,11,12]. However, the factors and mechanisms that influence this behavior in Brandt’s voles have yet to be determined. Additionally, in a certain percentage of captive Brandt’s voles strong food grinding was observed, with the ground chow comprising up to 12% of their body mass [17], compared to mice, with orts production representing only 3~4% of their body mass [1]. As a result, we propose that captive Brandt’s voles serve as a natural and viable model for researching the factors and mechanisms underlying food-wasting behavior, ultimately aiding in the conservation of grassland plants. In our previous study [17], we postulated that the gut microbiota might be associated with food grinding and demonstrated a significant correlation between the relative abundances of gut microbiome components and their metabolites with food grinding in captive Brandt’s voles. Nevertheless, the precise role of gut microbiota in food grinding remains poorly understood.

In this study, we reduced the food supply and investigated the response of voles exhibiting high-level food grinding to the reduction in the food supply. Specially, we assessed the variations in their body mass, consumed and ground food, ground-to-consumed food ratio, fecal short-chain fatty acids (SCFAs), fecal microbial alpha and beta diversity, and the predicted functions of the fecal microbiota, as well as the correlations among these variables. The ultimate aim was to test the effect of food supply variations on food grinding and further determine the relation between the gut microbiota and this behavior.

## 2. Materials and Methods

### 2.1. Animals and Housing

The environmental conditions for breeding Brandt’s voles (captured from the grasslands of Xilinhot, China) at the College of Bioscience and Biotechnology, Yangzhou University, China, were as follows: air temperature, 22 ± 1 °C; photoperiod, 12 L:12 D (light period: 06:00–18:00); and relative humidity, 50 ± 5%. All voles were offered water ad libitum and were freely fed rodent pellet chow (Yizheng Animal Biotechnology Co., Ltd., Yangzhou, China). At 21 days of age, young voles were weaned and separately housed in plastic cages until they were 60 days old, which represented the adult stage for this species as 8 weeks was considered as the age of sexual maturation [12,18]. The food ground by each vole, from 21 to 60 days of age, was recorded. 

### 2.2. Experimental Design

At 60 days of age, 23 male Brandt’s voles, including voles that ground both large and small amounts of food according to the record, were randomly chosen to perform the 15 g daily food supply experiment for 2 weeks. An iron mesh, with holes measuring 6 × 6 mm, was positioned within each cage, situated 1 cm above the cage floor. This mesh effectively filtered the ground chow and ensured that the voles were unable to ingest it. Of the 23 male voles, 6 that ground a higher average daily amount of food were marked as the high-level food-grinding voles based on the upper 27% rule [19]. These six voles made up the 15 g-food supply group. Subsequently, we conducted a food restriction experiment on this group of voles, wherein the supply of chow was reduced from 15 g to 7.5 g for a duration of 2 weeks. This new amount of chow provided was below the average daily intake (8~10 g) observed in our previous study [17]. Consequently, the voles involved in the food restriction experiment were classified as the 7.5 g-food supply group. 

Daily amounts of consumed and ground food were measured, and body mass measurements were conducted every 3 days in both the 15 g- and 7.5 g-food supply groups. Fresh fecal pellets were collected from each vole twice during the last week of each experiment and placed in sterile tubes, which were then immediately frozen in liquid nitrogen and stored at −70 °C. The daily ground food was dried in an oven set at 40 °C, to a constant weight, and then weighed. Calculations of daily food consumption were conducted as described by Shen et al. [17]. The average daily amounts of consumed and ground food for each experiment were calculated separately. Relative food consumption and ground food, and the ground to consumed food ratio were calculated as described by Shen et al. [17]. The change in relative ground food was obtained as the difference between the relative ground food in the 7.5 g-food supply group and that in the 15 g-food supply group divided by that in the 15 g-food supply group. The same formula was used to calculate the change in relative food consumption. The body mass growth rate was determined by calculating the difference in body mass between day 14 and day 1, divided by the body mass on day 1, for both the 15 g- and 7.5 g-food supply experiments. All protocol and procedures were approved by the Animal Care and Use Committee of the Faculty of Veterinary Medicine at the Yangzhou University (No. NSFC2020-SKXY-6).

#### 2.2.1. 16S rRNA Gene Sequencing and Bioinformatic Analysis of Fecal Microbiota

Two fecal samples collected during the second week of each experiment from each vole were mixed together as the fecal sample of the individual and then sent to Novogene Co. Ltd., Beijing, China, for DNA extraction and 16S rRNA gene sequencing of the fecal microbiota, according to the procedures of Dai et al. [20]. Briefly, the V4 distinct region of the prokaryotic 16S rRNA gene was amplified used specific primers (515F: 5′-GTGCCAGCMGCCGCGGTAA-3′; 806R: 5′-GGACTACHVGGGTWTCTAAT-3′). Paired-end reads were assigned to each sample referring to their unique barcodes and truncated by cutting off the barcodes and primer sequence. Paired-end reads were spliced using FLASH (V 1.2.7) to obtain the raw tags. The raw tags were filtered by QIIME (V1.9.1) and compared with the Silva database [21] using the UCHIME algorithm [22] to remove the chimera sequences, and finally obtain the effective tags. The Uparse software (v7.0.1001) was used to analyze the sequences [23]. The sequence pairs were identified as different bacterial operational taxonomic units (OTUs) according to a minimum identity threshold of 97%. Taxonomic information relating to operational taxonomic unit (OTU) sequences was annotated using the 11_4 release of the RDP database, following the methodology of the Mothur algorithm (Threshold: 0.8–1) [24]. The sequence data can be accessed at the NIH Sequence Read Archive under the Bioproject ID PRJNA868862. Alpha and beta diversity analyses were conducted as outlined in the study by Dai et al. [20]. To determine the alpha diversity of the fecal microbiota, the observed species (OBSP), Chao1, abundance-based coverage estimator (ACE), and Shannon and Simpson indices were employed. The calculation of all indices was performed using Quantitative Insights into Microbial Ecology (QIIME, Version 1.7.0). Dominant biomarkers in the fecal microbiota of the groups receiving 15 g and 7.5 g of food were identified utilizing linear discriminant analysis (LDA) effect size (LEfSe; LDA score > 3) with the online LEfSe program (http://huttenhower.sph.harvard.edu/galaxy/, accessed on 23 July 2022) [20].

#### 2.2.2. Kyoto Encyclopedia of Genes and Genomes (KEGG) Pathway Prediction of Fecal Microbiota 

The OTUs were clustered using QIIME 2 (version 2023.5) referring to the Greengene data resource (version GG 13.5) [25] based on the OTU data of the 16S rRNA gene sequencing. Then, the biological metagenome functions of the fecal microbiota of voles for the 15 g- and 7.5 g-food supply groups were predicted using the PICRUSt program with the type of KEGG Orthologs (http://huttenhower.sph.harvard.edu/galaxy/, accessed on 25 July 2022), according to the information of clustered OTUs, and annotated using the KEGG pathway database [20].

#### 2.2.3. SCFA Assay

The content of seven SCFAs in the feces of voles, including acetate, propionate, isobutyrate, butyrate, isovalerate, valerate, and caproate, were identified and quantified by the Agilent Technologies (Böblingen, Germany) headspace gas chromatography system (Agilent 7890A-7697A) based on a protocol detailed by Shen et al. [17]. 

#### 2.2.4. Statistical Analysis

Shapiro–Wilk and Levene tests were used to examine the normality and homogeneity of variance in the data, respectively. We used the non-parametric Mann–Whitney U test or *t* test in SPSS Statistics (version 22; IBM Corp., Armonk, NY, USA) to investigate disparities among the variables, including the number of total tags, taxon tags and OTUs, enriched biomarkers, alpha diversity indices, SCFAs concentrations, enriched KEGG pathways, body mass, body mass growth rate, relative food consumption, relative ground food, and ratios of ground-to-consumed food in food-grinding voles between the 15 g- and 7.5 g-food supply groups, as well as the differences between changes in the relative ground food and relative food consumption. If the data of variables exhibited a normal distribution and homogeneity of variance, we used the T test; otherwise, the non-parametric Mann–Whitney U test was used. Variations in the beta diversity of the fecal microbiota were tested by performing permutational multivariate analysis of variance (PERMANOVA), with Bray–Curtis distance matrix analysis conducted using the nested adonis function in the “vegan” package in R ver. 4.0.4. Spearman’s rank correlations were computed between variables that showed no significant variation and those that exhibited significant variation using R ver. 4.0.4. Correlations were considered significant when the false discovery rate (FDR) *p*-value was <0.05 [26]. The level of statistical significance was determined at *p* < 0.05.

## 3. Results

### 3.1. Differences in Food Intake, Food Ground, and Body Mass Growth Rate

The relative food consumption (*p* = 0.004) and ground food (*p* = 0.002), the ground to consumed food ratio (*p* < 0.001), and the body mass growth rate (*p* < 0.001) significantly decreased for the high-level food-grinding voles when the food supply was reduced from 15 to 7.5 g (Figure 1A–D). However, body weight did not sharply differ between the 15 g- and 7.5 g-food supply groups (*p* = 0.724; Figure 1E). The change in relative ground food was greater than the change in the relative food consumption (*p* = 0.002) when the food supply was reduced from 15 to 7.5 g (Figure 2).

### 3.2. OTUs Analysis

A total of 18 phyla, 28 classes, 44 orders, 94 families, 175 genera, and 2098 OTUs were identified in both the 15 g- and 7.5 g-food supply groups. The dominant phyla identified were Bacteroidetes (43.97%), followed by Firmicutes (42.69%), Proteobacteria (4.66%), and Actinobacteria (3.28%) (Figure 3A). The dominant genera identified were *Barnesiella* (14.10%), followed by *IV* (7.57%), *Staphylococcus* (3.09%), *Atopobium* (2.10%), and *Allobaculum* (1.74%) (Figure 3B). The values of the average Good’s coverage were as high as 99.75%, indicating that the16S rDNA sequencing method could identify the vast majority of taxa among the microbiota presenting in the fecal samples (Table 1). The numbers of total tags, taxon tags, and OTUs for the 7.5 g-food supply group were not significantly different from those for the 15 g-food supply group (*p* = 0.174, 0.852, and 0.076, respectively; Table 1).

### 3.3. Differences in Alpha and Beta Diversities of Fecal Microbial Community

No obvious differences between the two food supply groups were detected with respect to the Chao1, Shannon, Simpson and the ACE indices, and OBSP values (*p* = 0.135, 0.367, 0.485, 0.094, and 0.069, respectively; Figure 4A–E). The beta diversity of the fecal microbiota did not sharply vary after the food supply was reduced to 7.5 g (*F =* 1.070, *p* = 0.34).

### 3.4. Differences in the Abundances of the Enriched Biomarkers of Fecal Microbial Community

The biomarkers for the Bacilli class, Bacillales and Lactobacillales orders, Staphylococcaceae and Aerococcaceae families, and *Staphylococcus*, *Aerococcus*, and *Jeotgalicoccus* genera (Figure 5A,B) were enriched for the 15 g-food supply group. The biomarkers for the Flavobacteriia and Betaproteobacteria classes, Flavobacteriales and Burkholderiales orders, Flavobacteriaceae family, and *Un--s-Clostridiaceae bacterium GM1* genus (Figure 5A,B) were enriched for the 7.5 g-food supply group. The relative abundances of Bacilli (*p* = 0.009), Bacillales (*p* = 0.009), Lactobacillales (*p* = 0.026), Staphylococcaceae (*p* = 0.015), Aerococcaceae (*p* = 0.009), *Staphylococcus* (*p* = 0.026), *Aerococcus* (*p* = 0.009), and *Jeotgalicoccus* (*p* = 0.009) for the 15 g-food supply group were all higher than those for the 7.5 g-food supply group (Figure 6A–H). Moreover, the relative abundances of Flavobacteriia (*p* = 0.015), Betaproteobacteria (*p* = 0.026), Flavobacteriales (*p* = 0.015), Burkholderiales (*p* = 0.026), Flavobacteriaceae (*p* = 0.015), and *Un--s-Clostridiaceae bacterium GM1* (*p* = 0.026) were all higher for the 7.5 g-food supply group than for the 15 g-food supply group (Figure 6I–N). 

### 3.5. The Enriched KEGG Pathways of Fecal Microbial Community and Differences in the Content of Fecal SCFAs

The KEGG pathways associated with *Staphylococcus aureus* infection, the ubiquitin system, and alpha-linolenic acid metabolism (*p* = 0.026, 0.041, and 0.026, respectively) were more enriched for the 15 g-food supply group compared to that for the 7.5 g-food supply group (Figure 7). Further, the acetate content was lower for the 7.5 g-food supply group than for the 15 g-food supply group (*p* = 0.004; Figure 8A). Meanwhile, the propionate, isobutyrate, butyrate, isovalerate, valerate, and caproate content did not significantly differ between the 15 g- and 7.5 g-food supply groups (*p* = 0.159, 0.681, 0.818, 0.872, 0.485, and 0.065, respectively; Figure 8B–G).

### 3.6. Spearman’s Rank Correlations

Propionate content was positively correlated with relative ground food, ratio of ground to consumed food, and acetate content (*p* = 0.033, 0.024, and 0.016, respectively; Figure 9). Caproate content was positively correlated with relative food consumption and ground food (*p* = 0.012 and 0.042, respectively), acetate content (*p* < 0.001), and pathway enrichment of alpha-linolenic acid metabolism (*p* = 0.023), but negatively correlated with a relative abundance of Betaproteobacteria class and Burkholderiales order (*p* = 0.008 and 0.011, respectively; Figure 9). Isovalerate and valerate content were both negatively correlated with a relative abundance of Betaproteobacteria class (*p* = 0.028 and 0.003, respectively; Figure 9) and Burkholderiales order (*p* = 0.035 and 0.003, respectively). OBSP was positively correlated with a relative abundance of Betaproteobacteria class and Burkholderiales order (*p* = 0.007 and 0.011, respectively; Figure 9). Chao1 and ACE were both positively correlated with a relative abundance of Betaproteobacteria class (*p* = 0.039 and 0.042, respectively; Figure 9).

## 4. Discussion

In this study, we investigated the effect of food supply changes on food grinding in male Brandt’s voles and its potential underlying mechanisms with respect to the gut microbiota. This study lays the groundwork for future investigations into this behavior in a wider array of wild voles, particularly those coexisting in this palearctic region, in order to reveal the impact of gut microbiota on such behavior. When the food supply for Brandt’s voles was restricted to 7.5 g, which was below the average daily food intake, the body mass growth rate, food consumption, amount of ground food, and the ground-to-consumed food ratio decreased. This indicates that food restriction (7.5 g) can reduce food consumption and restrict vole growth. The abundances of certain fecal microbiota, as well as the levels of specific short-chain fatty acids (SCFAs), exhibited variation; however, there was no significant alteration observed in alpha diversity indices or the structure of the fecal microbial community following food restriction. These findings suggest that while the reduction in food supply led to a sharp decrease in food grinding behavior, it did not induce substantial changes in the gut microbiota on a large scale. Consistently, food restriction (80% the free-fed food intake) did not alter the structure of the gut microbiota in Brandt’s voles [27]. The most abundant phyla were Firmicutes, Bacteroidetes, and Proteobacteria, which is consistent with the gut microbial community of Brandt’s voles studied by Xu et al. [28]. In line with our earlier investigation comparing groups with differing degrees of food grinding, there was no significant difference in alpha diversity, with only limited dissimilarity observed in beta diversity [17]. We hypothesize that substantial alterations in gut microbiota diversity do not coincide with the occurrence of food grinding. Indeed, maintaining a stable gut microbiota is highly advantageous for the host organisms [29,30].

Although body mass did not differ remarkably between the 15 g- and 7.5 g-supply groups when the food supply was reduced to 7.5 g, food consumption decreased, and the body mass growth rate was negative and significantly lower than that observed when the food supply was 15 g. Consistently, the Brandt’s voles lost body mass when they were restricted of food in both warm and cold conditions [27], and food restriction induced a significant decrease in body mass in the striped hamster (*Cricetulus barabensis*) [9]. The reason behind the body weight not being significantly different between the 15 g- and 7.5 g-food supply groups is that for the 15 g-food supply group, the body weights of voles increased from day 1 to day 14, whereas for the 7.5 g-food supply group, the body weights decreased from day 1 to day 14, and the food restriction experiment was conducted immediately following the 15 g-food supply experiment. Thus, the mean body weights were not obviously different between the 15 g- and 7.5 g-food supply groups. 

The ratio of ground to consumed food decreased sharply, consistent with the higher amplitude of variation in relative ground food than in relative food consumption, indicating that the quantity of ground food decreased more than food consumption under a reduced food supply. Ort production in mice decreases to zero when they are offered 80% of their food intake [1]. Therefore, we concluded that the food grinding could be influenced by the quantity of food supplied. This suggests that when vegetation in grasslands is plentiful, food waste by wild Brandt’s voles is more extensive. We propose that rodents should cause more severe damage to plants or food by engaging in this behavior when food or vegetation is abundant. This may help to partially explain the enormous influence that rodents have on the dynamics of ecosystems by the alteration of plant–herbivore interactions, which are triggered by the increase in the vegetation brought about by environmental or climatic changes [31]. Moreover, the inhibitive effect of food restrictions on food grinding could be explained by the hypothesis of an optimal food intake strategy [32] with food components having high energy per gram selected through grinding, to render the energy intake as large as possible [1,5]. When the food supply was reduced to less than the average food intake and the body mass growth rate was negative, food grinding would further reduce the available food to levels which could not meet the daily energy demand. Additionally, food grinding is an energy-cost behavior. Thus, no advantage would be gained regarding food selectivity or food grinding. Small mammals can regulate the energy budget in response to the decrease in food supply to sustain periods of food shortage [9]. Therefore, voles would reduce their food grinding and ingest as much food as possible to maximize energy intake and save energy expenditure. In our previous study, no significant correlation was detected between food consumption and relative ground food [17], whereas in this study, a simultaneous decline in both food consumption and ground food was observed. This inconsistency might be attributed to the negative effects of food restriction on food consumption and grinding. The growth rate of body mass declined in tandem with the reduction in the relative amount of ground food and the ground-to-consumed food ratio. This implies that food grinding potentially offers additional advantages for voles beyond the mere maximization of energy intake. However, further investigation is necessary to fully comprehend these potential benefits. 

In this study, the acetate content was also lower for the 7.5 g-food supply group than for the 15 g-food supply group, supporting our speculation that acetate could promote food grinding [17]. Therefore, the decreased acetate content might have restrained food grinding in this study. The concentration of caproate was found to positively correlate with the relative amount of food ground, while exhibiting a decreasing trend (*p* = 0.065) following food restriction. This suggests that caproate may also contribute to the regulation of food grinding. The genera *Aerococcus* and *Un--s-Clostridiaceae bacterium GM1* were the common biomarkers observed, based on two LEfSe analyses, that could differentiate the high- and low-level food-grinding groups in our previous study [17], as well as the 15 g- and 7.5 g-food supply groups in this study. This indicates a strong relationship between the *Aerococcus* or *Un--s-Clostridiaceae bacterium GM1* genera and food grinding, which supports our previous speculation that gut microbiota participate in the regulation of food grinding [17]. It is likely that these two genera could contribute to the regulation of food grinding, with the genus *Un--s-Clostridiaceae bacterium GM1* inhibiting and the genus *Aerococcus* promoting it. The decreased abundance of *Aerococcus* and increased abundance of *Un--s-Clostridiaceae bacterium GM1,* in conjunction with the decreased acetate content, suggest that the effect of these two genera on food grinding may be caused by modifications in the acetate production. Therefore, we inferred that changes in the abundance of the gut microbiome components and their metabolite SCFAs due to food restriction would be helpful in inhibiting food grinding in Brandt’s voles.

Betaproteobacteria and Burkholderiales abundance in adult male Brandt’s vole increased along with the decrease in body mass when food supply was restricted [27]. In this study, abundances of Betaproteobacteria and Burkholderiales were consistently higher in the 7.5 g-food supply group. The rising trend of OBSP (*p* = 0.067) following food restriction should be caused by an increase in microorganisms in the Betaproteobacteria and Burkholderiales, according to the positive correlations found between OBSP and the relative abundance of these two groups of bacteria. Negative correlations between the relative abundances of Betaproteobacteria and Burkholderiales and the levels of caproate, isovalerate, and valerate suggest that these microbial taxa may inhibit the production of SCFAs within the gastrointestinal tract of Brandt’s vole. An increase in *Staphylococcus aureus* has been observed for obese individuals [33]. Further, reductions in the body weight, *Staphylococcus* abundance in the rectum, and Staphylococcaceae and *Jeotgalicoccus* abundance in the cecum have been observed for Kunming mice by the administration of copper [34]. In this study, the abundances of the genera *Staphylococcus* and *Jeotgalicoccus*, and family Staphylococcaceae, along with a decrease in the body mass, were consistently lower for the 7.5 g-food supply group. It is worth noting that the Kunming mice showed similar variation patterns in the abundance of Staphylococcaceae, *Staphylococcus*, and *Jeotgalicoccus* with Brandt’s voles, along with a reduction in body mass caused by either food restriction or copper treatment. Food grinding, on the other hand, was not described in those Kunming mice [34]. However, it is unclear if these point to copper-related associations with food grinding or just a similar gut microbial community between these two animals. Staphylococcaceae species are pathogens [35], and *S. aureus* infects a variety of tissues, organs, and systems in humans [36,37,38]. The pathway associated with *S. aureus* infection was more enriched for the 15 g-food supply group than for the 7.5 g-food supply group owing to the abundance of the genus *Staphylococcus* and the family Staphylococcaceae. Therefore, it is likely that food restriction can help reduce the abundance of pathogenic Staphylococcaceae species. Alpha-linolenic acid is an antibacterial substance that inhibits *S. aureus* [39,40]. Moreover, autophagy, mediated by ubiquitin receptors, is essential for zebrafish resistance to *S. aureus* [41]. Therefore, the enrichment of pathways related to the ubiquitin system and alpha-linolenic acid metabolism for the 15 g-food supply group could be associated with a resistance to *Staphylococcus* bacteria. In Rex rabbits with an increased weight, the concomitant enrichment of pathways related to alpha-linolenic acid metabolism, and increased *S. aureus* colonization among the fecal microbiota was observed [42]. *Jeotgalicoccus* is supposed to ferment carbohydrates or proteins [34] and to deliver potentially helpful bodily services [43]. Reducing food intake through dietary restrictions could lead to an inadequate amount of proteins or carbohydrates in the gut for fermentation, which would lower the abundance of *Jeotgalicoccus* in our study. In addition, the contrasting fluctuations in the abundance of taxa within the Bacilli and Flavobacteriia classes following food restriction suggest a potential mutual inhibition between these bacterial classes in Brandt’s voles.

In this study, differences in the amount of ground food, the ground-to-consumed food ratio, the abundances of fecal microbiota components, and the metabolite content between adult male voles with different food supplies were observed. Based on these findings, we speculated on the effects of food restriction on food grinding, as well as the correlation between the gut microbiota, in addition to their metabolites, and food grinding. This study, along with our previous research [17], preliminarily clarifies the factors influencing food grinding and the mechanisms underlying this characteristic from the perspective of the gut microbiota. However, we could not distinguish whether the direct effect of food restriction, owing to the insufficient energy supply, or indirect effect of food restriction, via the gut microbiota, on food grinding was predominantly important. Food restriction in association with gut microbiota transplantation experiments should help to address this ambiguity. Additionally, the precise gut microbiota-associated mechanisms underlying food grinding and the functions of food grinding in voles require further research. We deduce that in young Brandt’s voles, food grinding is probably less prominent than in adults based on the age-related rise in food grinding in mice [2] and our observation of the minimum occurrence of food grinding in immature Brandt’s voles during breeding (unpublished). Moreover, it is anticipated that among young Brandt’s voles, the inhibitory effect of food restriction on food grinding will be less pronounced. Although food grinding and gender in mice are uncorrelated [4], little is known about this activity in female Brandt’s voles. Further research is necessary since it is unclear how food restriction affects food grinding and what role gut microbiota plays in this behavior in female and young Brandt’s voles. To completely understand food grinding, more voles or rodents must be included in these studies.

## 5. Conclusions

Our study demonstrated that food restriction could reduce the amount of ground food to a greater extent than its effect on food consumption, and alter the abundance of the gut microbiome and its metabolites. The genera *Aerococcus* and *Un--s-Clostridiaceae bacterium GM1* could have a significant role in regulating food-grinding behavior, with acetate potentially serving as a crucial metabolite that influences this behavior. Thus, variations in the abundance of the gut microbiome components and their metabolite SCFAs, induced by food restriction, could contribute to the inhibition of food grinding in Brandt’s voles. This study suggests that reducing the food supply could effectively inhibit food grinding during laboratory rodent feeding, further supporting the hypothesis that food grinding could be motivated by an optimal food-intake strategy and partially verifying the hypothesis that the gut microbiota might be related to food grinding. These results could potentially provide guidance for reducing food waste during laboratory rodent maintenance, and promote deeper research on the gut microbial mechanism of food grinding and the potential role of food grinding. This will ultimately broaden our understanding of the effects of vegetation biomass and food abundance on food-wasting behavior in wild rodents.

## Figures and Tables

**Figure 1 animals-13-03424-f001:**
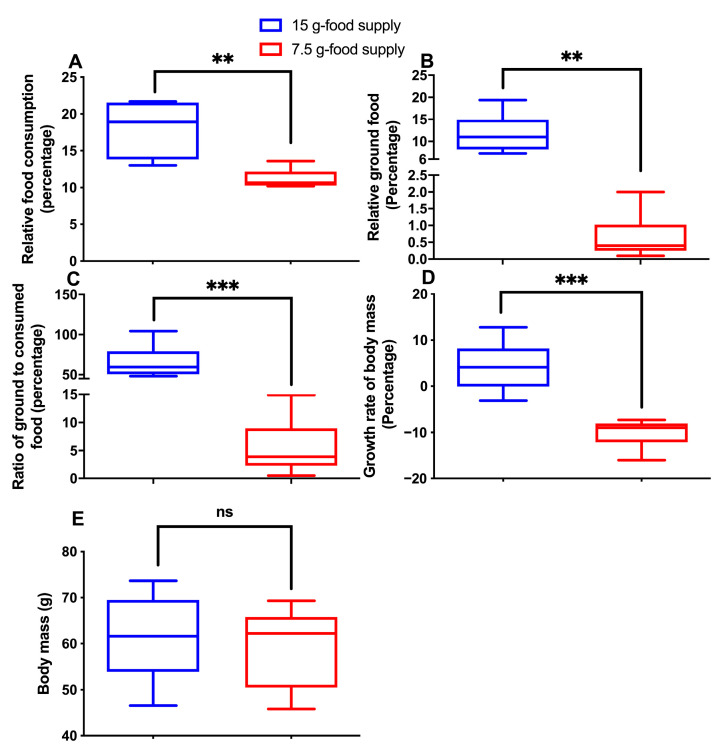
Differences in relative food consumption (**A**), relative ground food (**B**), food ground to consumption ratio (**C**), body mass (**D**), and growth rate of body mass (**E**) between the 15 g- and 7.5 g-food supply groups of Brandt’s voles. Shown in the box diagram are the lower range, the first quartile, the median, the third quartile, and upper range (*n* = 6). Note: ** and *** mean *p*-value was < 0.01 and 0.001, respectively; ns means not significant.

**Figure 2 animals-13-03424-f002:**
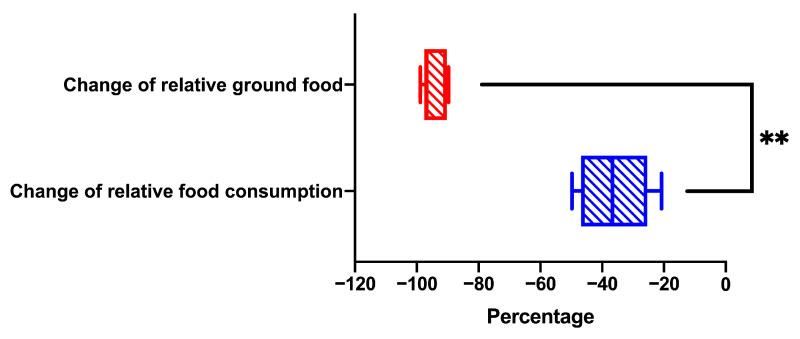
Differences between the changes of relative food consumption and relative ground food in Brandt’s vole when food supply reduced from 15 g to 7.5 g. Shown in the box diagram are the lower range, the first quartile, the median, the third quartile, and upper range (*n* = 6). Note: ** means *p*-value was <0.01.

**Figure 3 animals-13-03424-f003:**
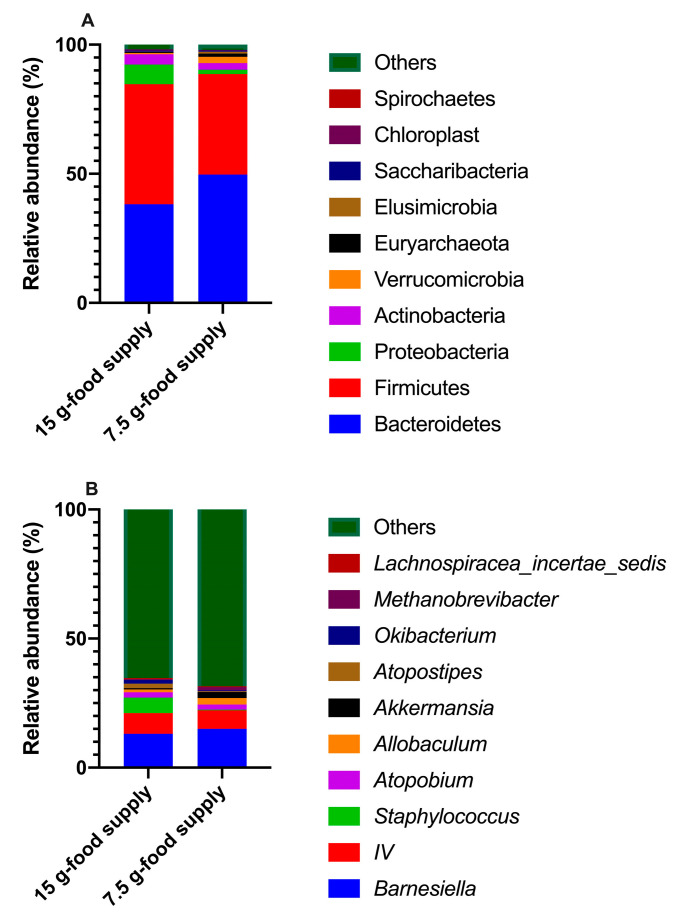
The components of top 10 operational taxonomic units at level of phylum (**A**) and genus (**B**) in both 15 g- and 7.5 g-food supply groups.

**Figure 4 animals-13-03424-f004:**
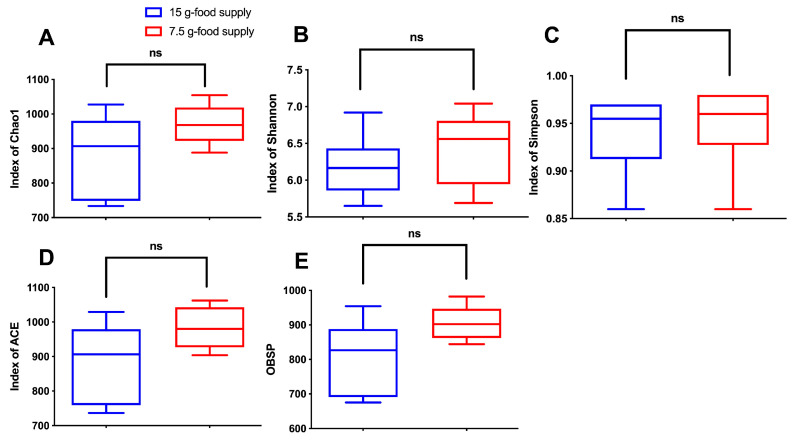
Differences in the fecal microbiota composition between the 15 g- and 7.5 g-food supply groups of Brandt’s voles. (**A**) Chao1; (**B**) Shannon index; (**C**) Simpson index; (**D**) abundance-based coverage estimator (ACE); (**E**) observed species (OBSP). Shown in the box diagram are the lower range, the first quartile, the median, the third quartile, and upper range (*n* = 6). Note: ns means not significant.

**Figure 5 animals-13-03424-f005:**
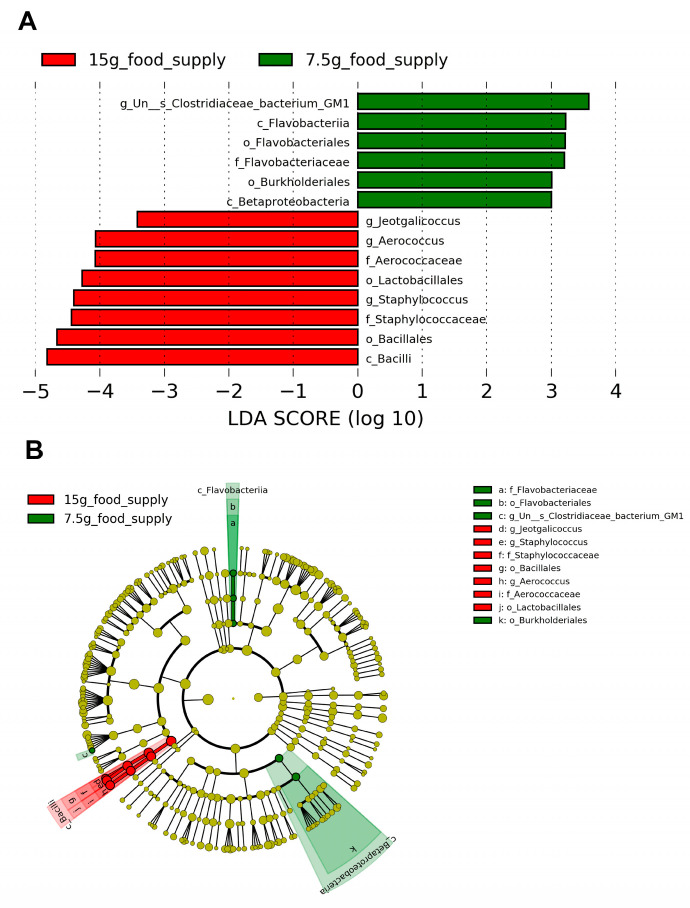
Biomarker of taxa with statistically significant differences in relative abundance selected in the fecal microbiota between the 15 g- and 7.5 g-food supply groups of Brandt’s voles (*n* = 6). (**A**) Histogram displaying the differential taxa using the linear discriminant analysis (LDA) effect size (LEfSe) analysis with an LDA score significant threshold > 3; (**B**) Cladogram displaying the phylogenetic position of differential taxa.

**Figure 6 animals-13-03424-f006:**
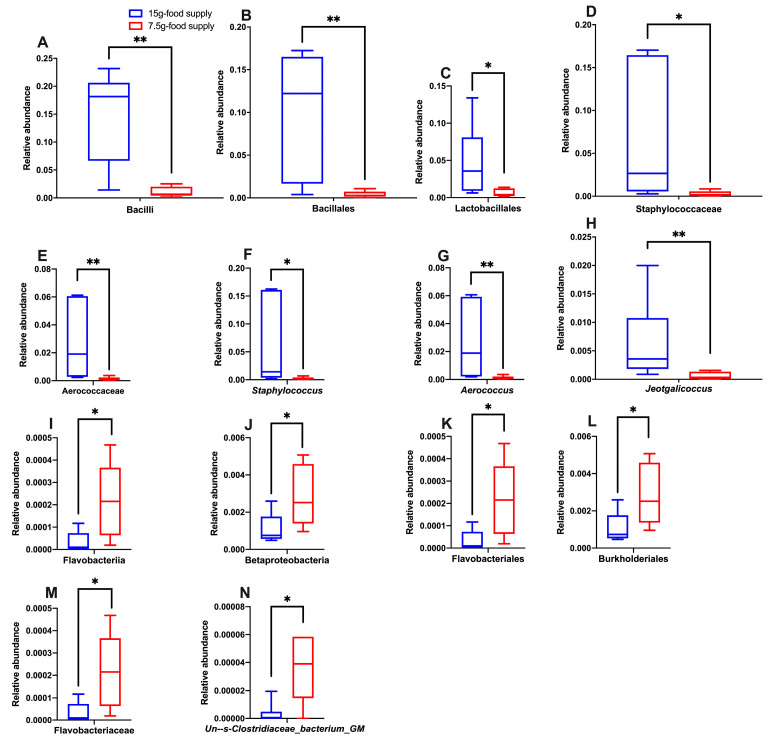
Differences in the relative abundances of fecal microbial taxa between the 15 g- and 7.5 g-food supply groups of Brandt’s voles. Bacilli (**A**), Bacillales (**B**), Lactobacillales (**C**), Staphylococcaceae (**D**), Aerococcaceae (**E**), *Staphylococcus* (**F**), *Aerococcus* (**G**), and *Jeotgalicoccus* (**H**), Flavobacteriia (**I**), Betaproteobacteria (**J**), Flavobacteriales (**K**), Burkholderiales (**L**), Flavobacteriaceae (**M**), and *Un--s-Clostridiaceae bacterium GM1* (**N**). Shown in the box diagram are the lower range, the first quartile, the median, the third quartile, and upper range (*n* = 6). Note: * and ** mean *p*-value was < 0.05 and 0.01, respectively.

**Figure 7 animals-13-03424-f007:**
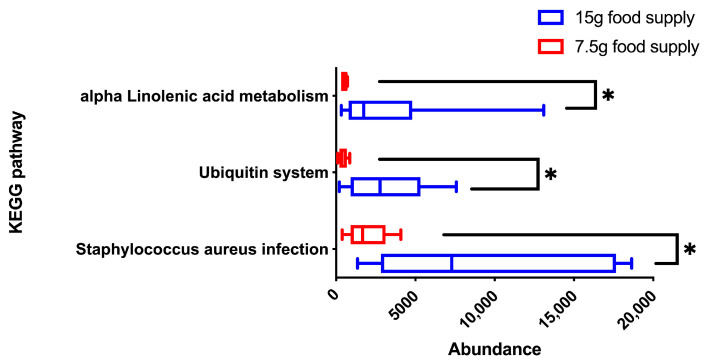
Differences in the abundances of fecal microbial KEGG pathways between the 15 g- and 7.5 g-food supply groups of Brandt’s voles. Shown in the box diagram are the lower range, the first quartile, the median, the third quartile, and upper range (*n* = 6). Note: * means *p*-value was < 0.05.

**Figure 8 animals-13-03424-f008:**
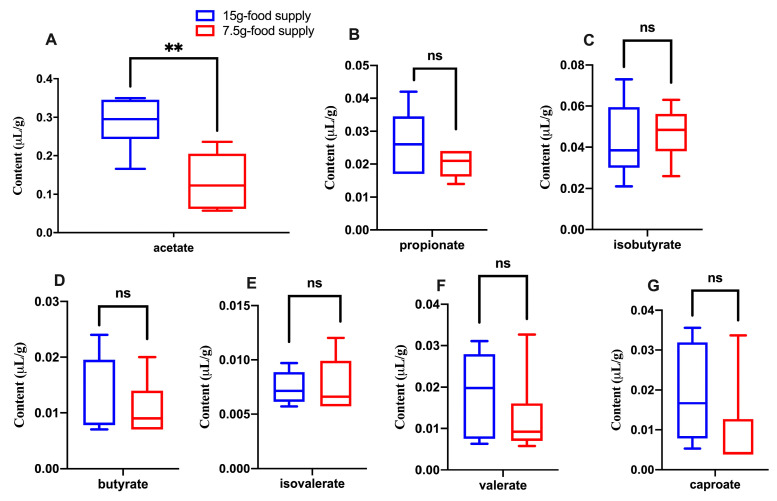
Difference of fecal short-chain fatty acids (SCFAs: acetate (**A**), propionate (**B**), isobutyrate (**C**), butyrate (**D**), isovalerate (**E**), valerate (**F**), and caproate (**G**)) content between the 15 g- and 7.5 g-food supply groups of Brandt’s voles. Shown in the box diagram are the lower range, the first quartile, the median, the third quartile, and upper range (*n* = 6). Note: ** means *p*-value was < 0.01; ns means not significant.

**Figure 9 animals-13-03424-f009:**
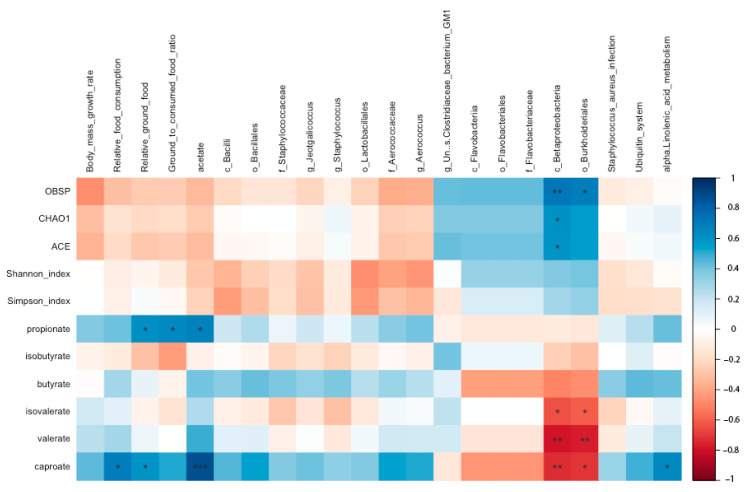
Correlations among variables using Spearman’s rank in the 15 g- and 7.5 g-food supply groups of Brandt’s voles. Note: *, **, and *** mean *p*-value was < 0.05, 0.01, and 0.001, respectively (*n* = 12).

**Table 1 animals-13-03424-t001:** Differences in total tags, taxon tags, operational taxonomic units (OTUs), and Good’s coverage in 16S rRNA libraries of fecal microbiota between the 15 g- and 7.5 g-food supply groups of Brandt’s voles (mean ± SE) (*n* = 6).

Groups	Total Tags	Taxon Tags	OTU Numbers	Goods Coverage
15 g-food supply	63,501 ± 1070	57,234 ± 1017	877 ± 47	99.77 ± 0.02
7.5 g-food supply	66,282 ± 1567	57,704 ± 2233	981 ± 22	99.75 ± 0.02

## Data Availability

The sequence data are available at the NIH Sequence Read Archive with Bioproject ID PRJNA868862.

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
