# Peer review of "Effect of Food Restriction on Food Grinding in Brandt’s Voles"

_animals, 2023, doi:10.3390/ani13213424_

Round 1

Reviewer 1 Report

Comments and Suggestions for Authors

This is a rather small-sized paper, only investigating 23 male subjects. Usually, the preferred format is Short Communication in top journals at least. Among the Pro’s, voles are an important taxonomic group for two reasons: firstly, they are major components of several ecosystem including endangered ones. Secondly, some vole species have been exerciting a relevant experimental role, especially in behavioral neurosciences.

Introduction

Lines 33-34

(1,2) or a behavior of wild herbivorous rodents involving cutting and discarding substantial amounts of grass or herbaceous plants material (3.3)

The range of different foods consumed by voles in general is likely more “omnivorous”, especially in the lactating period of females. Some more information on the dietary habits of these particular species are needed, since the paper should be restructured, in order to provide information about the ecology, the niches and more in general the ecosystems in which this unrelated vole species exert regulatory role.

Materials and Methods

Experimental Design 2.2

Lines 75-77

At 60 days of age, 23 male voles, including voles that ground both large and small amounts of food according to the record, were randomly chosen to perform the 15 g-daily food supply experiment for 2 weeks.

Some more information are necessary: at which age these animals generally reach sexual maturation. Which is average food intake per day recorded for these particular species? In the discussion it could be advisable to mention how much the present results can be similar, or different, in the case of both female and immature subjects.

2.2.2. Lines 121-122

Kyoto Encyclopedia of Genes and Genomes (KEGG) pathway prediction of fecal microbiota

This encyclopedia needs to be better explained. Especially, it is relevant to mention other similar genes/genomes data storage. A paragraph of the Methods should be devoted to this issue.

4. Discussion

Lines 263-264

In this study, we investigated the effect of food …gut microbiota

How such a statement can be extended to other vole species, eg those distributed in the Paleartic region?

Lines 336-338

An increase in S. aureus has been observed for obese individuals (21). Further, reductions in the body weight, Staphylococcus abundance in the rectum, and Staphylococcaceae levels in the cecum have been observed for Kunming mice (22).

What about other rodent species and which is the evolutionary or ecological significance of such a comparative difference between the present species and Kumning mice?

5. Conclusions

Lines 370-373

The present conclusion is very, very poor, also the discussion should be consistently extended. The present data in fact can be discussed in the framework of the role exerted by those rodent species as major components of different ecosystems. Starting from a few subjects of single species, some more general trends can be imagined and thoroughly discussed, given the very rich literature on the role exerted by rodents in general and voles in particular in ecosystemic regulations

Comments on the Quality of English Language

Average better to review.

Author Response

Dear  reviewers,

We are very grateful to you  for your critical comments and thoughtful suggestions concerning our manuscript entitled “Effect of Food Restriction on Food Grinding in Brandt’s Voles”(ID:animals-2617770). Based on these comments and suggestions, we have made careful modifications to the original manuscript. All changes made to the text according to the comments are in red and highlighted so that they may be easily identified. We hope it has reached your magazine’s standard. All of your questions were answered below.

Question 1. Introduction. Lines 33-34:(1,2) or a behavior of wild herbivorous rodents involving cutting and discarding substantial amounts of grass or herbaceous plants material (3.3).The range of different foods consumed by voles in general is likely more “omnivorous”, especially in the lactating period of females. Some more information on the dietary habits of these particular species are needed, since the paper should be restructured, in order to provide information about the ecology, the niches and more in general the ecosystems in which this unrelated vole species exert regulatory role.

Response: According to reviewer’s advice, we have added some ecological information about Brandt’s vole as “Brandt's voles (Lasiopodomys brandtii), a small herbivorous mammalian species, inhabit the grasslands of Inner Mongolia, China [10-13]. This species consumes a diverse array of herbaceous plants, displaying a particular preference for Leymus chinensis, Medicago varia, Stipa krylovii, and Saussurea runcinata [14,15]. Notably, wild Brandt's voles engage in wasteful behavior, gnawing on grasses without actually consuming them in the field. Remarkably, the quantity of grass wasted through non-feeding activity (excluding nesting material) accounts for a quarter of their daily food intake [16]. These feeding patterns have been shown to adversely impact the grassland ecosystems and livestock husbandry within the region [10-12].” in line 47 to 55. Brandt's voles are herbivorous mammalian species.

Question 2. Materials and Methods Experimental Design 2.2, Lines 75-77: At 60 days of age, 23 male voles, including voles that ground both large and small amounts of food according to the record, were randomly chosen to perform the 15 g-daily food supply experiment for 2 weeks. Some more information are necessary: at which age these animals generally reach sexual maturation. Which is average food intake per day recorded for these particular species? In the discussion it could be advisable to mention how much the present results can be similar, or different, in the case of both female and immature subjects.

Response: According to reviewer’s advice, we have added some information about the sexual maturation of  Brandt’s vole as “for 8 weeks was considered as the age of sexual maturation[12,18] ”in line 84 , and some information about the amount of daily food intake as “This new amount of chow provided was below the average daily intake (8~10 g) observed in our previous study[17].” in line 96 to 97. We also have added the discussion about the female and young voles as “We deduce that in young Brandt's voles, food grinding is probably less prominent than in adults based on the age-related rise in food grinding in mice [2] and our observation of the minimum occurrence of food grinding in immature Brandt's voles during breeding (unpublished). Moreover, it is anticipated that among young Brandt’s vole, the inhibitory effect of food restriction on food grinding will be less pronounced. Although food grinding and gender in mice are uncorrelated[4], little is known about this activity in female Brandt's voles. Further research is necessary since it is unclear how food restriction affects food grinding and what role gut microbiota plays in this behavior in female and young Brandt's voles. To completely understand food grinding, more voles or rodents must be included in these studies.” in line 438 to 447.

Question 3. Lines 121-122:Kyoto Encyclopedia of Genes and Genomes (KEGG) pathway prediction of fecal microbiota. This encyclopedia needs to be better explained. Especially, it is relevant to mention other similar genes/genomes data storage. A paragraph of the Methods should be devoted to this issue.

Response: The method of prediction of KEGG way is described in line 144 to 149 as “The OTUs were clustered using QIIME 2 (version 2023.5) referring to the Greengene data-resource (version GG 13.5) [25] based on the OTU data of the 16S rRNA gene sequencing. Then, the biological metagenome functions of the fecal microbiota of voles for the 15-and 7.5 g-food supply groups were predicted using the PICRUSt program with the type of KEGG Orthologs (http://huttenhower.sph.harvard.edu/galaxy/), according to the information of clustered OTUs, and annotated using the KEGG pathway database[20]”.

Question 4. Discussion. Lines 263-264. In this study, we investigated the effect of food …gut microbiota. How such a statement can be extended to other vole species, eg those distributed in the Palearctic region?

 Response: According to reviewer’s advice, we have added some statements in line 308 to 310 as “This study lays the groundwork for future investigations into this behavior in a wider array of wild voles, particularly those coexisting in this Palearctic region, in order to reveal the impact of gut microbiota on such behavior.” to extend to other vole species.

Question 5. Lines 336-338: An increase in S. aureus has been observed for obese individuals (21). Further, reductions in the body weight, Staphylococcus abundance in the rectum, and Staphylococcaceae levels in the cecum have been observed for Kunming mice (22). What about other rodent species and which is the evolutionary or ecological significance of such a comparative difference between the present species and Kumning mice?

Response: The variations of those microbiota were not found in other rodent species via the documents searching. In this part, we stated that the change of these microbiota in Brandt’s vole was similar to Kumning mice when body mass decreased, not different to Kumning mice. We have revised this part as “An increase in Staphylococcus aureus has been observed for obese individuals [32]. Further, reductions in the body weight, Staphylococcus abundance in the rectum,and Staphylococcaceae and Jeotgalicoccus abundance in the cecum have been observed for Kunming mice by the administration of copper[33]. In this study, the abundances of the genera Staphylococcus and Jeotgalicoccus, and family Staphylococcaceae, along with a decrease in the body mass, was consistently lower for the 7.5 g-food supply group. It's worth noting that the Kunming mice showed similar variation patterns in the abundance of Staphylococcaceae, Staphylococcus, and Jeotgalicoccus with Brandt's vole, along with a reduction in body mass caused by either food restriction or copper treatment. Food grinding, on the other hand, was not described in those Kunming mice [33]. However, it's unclear if these point to copper-related associations with food grinding or just a similar gut microbial community between these two animals.” in line 395 to 406 to discuss deeply.

Question 6. Conclusions. Lines 370-373: The present conclusion is very, very poor, also the discussion should be consistently extended. The present data in fact can be discussed in the framework of the role exerted by those rodent species as major components of different ecosystems. Starting from a few subjects of single species, some more general trends can be imagined and thoroughly discussed, given the very rich literature on the role exerted by rodents in general and voles in particular in ecosystemic regulations

Response: According to reviewer’s advice, we have revised the part of discussion as “This suggests that when vegetation in grasslands is plentiful, food waste by wild Brandt’s voles is more extensive. Similarly, other rodents should cause more severe damage to plants or food by engaging in this behavior when food or vegetation is abundant.” in line 349 to 352 and revised the conclusions as “This will ultimately broaden our understanding of the effects of vegetation biomass and food abundance on food-wasting behavior in wild rodents” in line 462 to 464, trying to extend the conclusions and discussion.

Reviewer 2 Report

Comments and Suggestions for Authors

The manuscript entitled “Effect of Food Restriction on Food Grinding in Brandt’s Voles” by Xin Dai and colleagues, submitted to the Animals, presents results of a study on food-grinding behaviour in rodents. Authors studied not only this behaviour, but also a role of the gut microbiota in this process. I found the study interesting. Food-grinding is poorly known behaviour; thus, such study is interesting and the results are valuable; however, I think that this version of manuscript could be improved.

General remarks
The experimental design could be presented in the better way, I believe. For example: (lines 75-77) “23 male voles, including voles that ground both large and small amounts of food according to the record, were randomly chosen to perform the 15 g-daily food supply experiment for 2 weeks.” (but how many animals were chosen to these groups?). Lines 79-81: “Of the 23 male voles, six that ground a higher average daily amount of food were marked as the high-level food-grinding group based on the upper 27% rule” and lines 81-84 “Thereafter, we conducted a food restriction experiment using the voles in this group, in which the chow supply was reduced from 15 to 7.5 g for 2 weeks, representing a final value below the average daily chow intake used for the 15 g-food supply experiment.” – how many experimental/control groups were used in this experiment?
Based on the table 1, there were two groups, each of six animals; however, 23 voles were used in the study.
Please forgive me for my possible misunderstanding, but I have problem to follow with the information in the 2. Materials and Methods section, especially 2.2. Experimental Design. I believe it could be difficult to understand for readers, as well.

Statistical analyses
Authors used the non-parametric Wilcoxon signed-rank test (lines 136-140) “to investigate the differences in the number of total tags, taxon tags and OTUs, OTU biomarkers, alpha diversity indices, SCFA contents, KEGG pathway enrichment, body mass, body mass growth rate, relative food consumption, relative ground food, and ground-to-consumed food ratios in food-grinding voles between the 15- and 7.5 g-food supply groups, as well as the differences between changes in the relative ground food and relative food consumption.”.
Thus, the first, many parameters were compared using this test.

For fecal microbiota analyses they performed permutational multivariate analysis of variance. However, also a numerous Spearman’s rank correlations among.

Wilcoxon signed-rank test and Spearman’s rank correlation are non-parametric methods, analysis of variance – parametric one. I have found no information why these methods were chosen, for example the information on checking assumptions for used the statistical methods etc. I am not sure, but when using so many Spearman’s rank correlations, Bonferroni correction should be used as well. See for example P values in the line 236 “P = 0.008, 0.041, 0.015, 0.050, 0.038, 0.011, 0.038, 0.035, 0.011, and 0.004”. If all of the values are statistically significant, if so many correlations were calculated using Spearman’s rank correlations? What more, I am not sure if it is good to perform so many correlation analyses, when n=6 (see for example figure 8).

On the Figures 1,2,3, 5,6,7 bars indicate standard errors are presented. However, standard errors are typical values presented for parametric tests results, not for non-parametric one (Wilcoxon signed-rank test is non-parametric). What more, if on the Figure 3 “Error bars indicate standard errors” are presented? They look rather like median, quartiles and range, I think. Check it please.

To sum up.
I have problem to understand the experimental design and I believe that more information on the statistical analyses are necessary – now, I am not sure, if good tests were chosen, and therefore if the conclusions are supported by the results (especially for the correlations analyses).

Figures
I believe that the Figures could be presented in a better way. Many of them are small, thus it is difficult to read them. When two groups are compared, the better/easier way to show statistical difference is using symbols like, for example, * – P < 0.05, ** – P < 0.01, *** – P < 0.001, ns – not significant (now, the differences are presented as “Different letters connect bars with significant differences” – I believe that is not the best way, when two groups are compared only; and it could be confusing for Figures 5-7).

I am not sure if I correctly understand the sentence (line 260) “Note: *, **, and *** mean false discovery rate-adjusted P-value was < 0.05, 0.01, and 0.001, respectively”. Check it, please.

Other comments:
Some parts of the manuscript could be rewritten in a better way, I think. For example, see the Introduction section.
- Line 40 “(Cameron and Speakman 2010)” should be “[1]” I think.
- Line 47 “In our previous study,” – I believe that it will be better to cite “[9]” here (i.e. “In our previous study [9], …”).
- The sentence (lines 47-52) “In our previous study, we first hypothesized that the gut microbiota could be related to food-grinding behavior and demonstrated that the relative abundances of gut microbiome components and their metabolites are significantly correlated with food-grinding behavior in captive Brandt’s voles (Lasiopodomys brandtii) [9], a small herbivorous mammalian species that exhibits food-wasting behavior, and is found in the grasslands of Inner 51 Mongolia, China [10-13].”
is really long sentence, and quite difficult to follow.
- Line 57 “The ultimate aim was to uncover the effect” – to ‘uncover’ or to ‘test’ (if there is any effect)?
- Lines 59-63 “Our findings provide guidance for reducing food waste during laboratory rodent rearing and provide new insights into the mechanisms through which the gut microbiota affect food-grinding behavior. This will ultimately broaden our understanding of the effects of vegetation biomass on food-wasting behavior in wild rodents.”
It is rather part of the Discussion section, I think.
- It should be used “Food grinding” phrase or “Food-grinding” phrase, in the section, and it the manuscript – not both of them (choose one of them, and use it consistently, please).

Author Response

Dear reviewers,

We are very grateful to you  for your critical comments and thoughtful suggestions concerning our manuscript entitled “Effect of Food Restriction on Food Grinding in Brandt’s Voles”(ID:animals-2617770). Based on these comments and suggestions, we have made careful modifications to the original manuscript. All changes made to the text according to the comments are in red and highlighted so that they may be easily identified. We hope it has reached your magazine’s standard. All of your questions were answered below.

Question 1.The experimental design could be presented in the better way, I believe. For example: (lines 75-77) “23 male voles, including voles that ground both large and small amounts of food according to the record, were randomly chosen to perform the 15 g-daily food supply experiment for 2 weeks.” (but how many animals were chosen to these groups?). Lines 79-81: “Of the 23 male voles, six that ground a higher average daily amount of food were marked as the high-level food-grinding group based on the upper 27% rule” and lines 81-84 “Thereafter, we conducted a food restriction experiment using the voles in this group, in which the chow supply was reduced from 15 to 7.5 g for 2 weeks, representing a final value below the average daily chow intake used for the 15 g-food supply experiment.” – how many experimental/control groups were used in this experiment? Based on the table 1, there were two groups, each of six animals; however, 23 voles were used in the study. Please forgive me for my possible misunderstanding, but I have problem to follow with the information in the 2. Materials and Methods section, especially 2.2. Experimental Design. I believe it could be difficult to understand for readers, as well.

Response: According to reviewer’s advices, we have revised this part to make it more clear as “Of the 23 male voles,six that ground a higher average daily amount of food were marked as the high-level food-grinding voles based on the upper 27% rule [19], as well as belonging to the 15 g-food supply group. Subsequently, we conducted a food restriction experiment on this group of voles, wherein the supply of chow was reduced from 15 g to 7.5 g for a duration of 2 weeks. This new amount of chow provided was below the average daily intake (8~10 g) observed in our previous study[17]. Consequently, the voles involved in the food restriction experiment were classified as the 7.5 g-food supply group.” in line 91 to 98.

Question 2. Statistical analyses. Authors used the non-parametric Wilcoxon signed-rank test (lines 136-140) “to investigate the differences in the number of total tags, taxon tags and OTUs, OTU biomarkers, alpha diversity indices, SCFA contents, KEGG pathway enrichment, body mass, body mass growth rate, relative food consumption, relative ground food, and ground-to-consumed food ratios in food-grinding voles between the 15- and 7.5 g-food supply groups, as well as the differences between changes in the relative ground food and relative food consumption.”. Thus, the first, many parameters were compared using this test. For fecal microbiota analyses they performed permutational multivariate analysis of variance. However, also a numerous Spearman’s rank correlations among. Wilcoxon signed-rank test and Spearman’s rank correlation are non-parametric methods, analysis of variance – parametric one. I have found no information why these methods were chosen, for example the information on checking assumptions for used the statistical methods etc. I am not sure, but when using so many Spearman’s rank correlations, Bonferroni correction should be used as well. See for example P values in the line 236 “P = 0.008, 0.041, 0.015, 0.050, 0.038, 0.011, 0.038, 0.035, 0.011, and 0.004”. If all of the values are statistically significant, if so many correlations were calculated using Spearman’s rank correlations? What more, I am not sure if it is good to perform so many correlation analyses, when n=6 (see for example figure 8).

Response: According to reviewer’s advices, we have added some information about how to choose the test method as “Shapiro–Wilk and Levene tests were used to examine the normality and homogeneity of variance of the data, respectively. We used the non-parametric Mann-Whitney U test or T test in SPSS Statistics (version 22; IBM Corp., Armonk, NY, USA) to investigate disparities among the variables, including the number of total tags, taxon tags and OTUs, enriched biomarkers, alpha diversity indices, SCFAs concentrations, enriched KEGG pathways, body mass, body mass growth rate, relative food consumption, relative ground food, and ratios of ground-to-consumed food in food-grinding voles between the 15- and 7.5 g-food supply groups, as well as the differences between changes in the relative ground food and relative food consumption. If the data of variables exhibited normal distribution and homogeneity of variance, we used the T test; otherwise, the non-parametric Mann-Whitney U test was used.” in line 156 to166. When we analyzed using Spearman’s rank correlations, the P value was adjusted by the method of FDR (false discovery rate). The method of FDR is one effective method of the P value correction used in the analysis of Spearman’s rank correlations referring to the study of Dovrou et al.(2023) (Dovrou A, Bei E, Sfakianakis S, Marias K, Papanikolaou N, Zervakis M. Synergies of Radiomics and Transcriptomics in Lung Cancer Diagnosis: A Pilot Study. Diagnostics (Basel). 2023,13(4):738. doi: 10.3390/diagnostics13040738). We chose Spearman’s rank correlations because data of many variables were not normally distributed and the sample size was small. We used Spearman’s rank correlations to find the potential correlations between these variables.

Question 3. On the Figures 1,2,3, 5,6,7 bars indicate standard errors are presented. However, standard errors are typical values presented for parametric tests results, not for non-parametric one (Wilcoxon signed-rank test is non-parametric). What more, if on the Figure 3 “Error bars indicate standard errors” are presented? They look rather like median, quartiles and range, I think. Check it please.

Response: According to reviewer’s suggestion, we have changed all of the bar graphs to box plots, and changed black and white graphs to color graphs. Yes, on the figure 3 it was not the error bar. We have corrected “Error bars indicate standard errors” in the caption as “Shown in the box diagram are the lower range, the first quartile, the median, the third quartile, and upper range” in line 234 to 235.

Question 4. To sum up. I have problem to understand the experimental design and I believe that more information on the statistical analyses are necessary – now, I am not sure, if good tests were chosen, and therefore if the conclusions are supported by the results (especially for the correlations analyses).

Response: According to reviewer’s advices, we have added more information about the statistical analyses. We also added the information about how to choose the method of statistical analyses. We consider the methods of statistical analyses are appropriate, which can support our conclusions.

Question 5. Figures. I believe that the Figures could be presented in a better way. Many of them are small, thus it is difficult to read them. When two groups are compared, the better/easier way to show statistical difference is using symbols like, for example, * – P < 0.05, ** – P < 0.01, *** – P < 0.001, ns – not significant (now, the differences are presented as “Different letters connect bars with significant differences” – I believe that is not the best way, when two groups are compared only; and it could be confusing for Figures 5-7). I am not sure if I correctly understand the sentence (line 260) “Note: *, **, and *** mean false discovery rate-adjusted P-value was < 0.05, 0.01, and 0.001, respectively”. Check it, please.

Response: According to reviewer’s suggestion, we have shown statistical difference using symbols in all figures, as “*, **, and *** mean P-value was < 0.05, 0.01, and 0.001, respectively; ns means not significant”. “Note: *, **, and *** mean false discovery rate-adjusted P-value was < 0.05, 0.01, and 0.001, respectively” is correct, because the P- value in the spearman’s rank correlation analysis was adjusted using method of false discovery rate. But we have modified this sentence to make it more concise as “Note: *, **, and *** mean P-value was < 0.05, 0.01, and 0.001, respectively” in line 303.

Question 6. Line 40 “(Cameron and Speakman 2010)” should be “[1]” I think.

Response: According to reviewer’s suggestion, we have corrected this mistake in line 39.

Question 7. Line 47 “In our previous study,” – I believe that it will be better to cite “[9]” here (i.e. “In our previous study [9], …”).

Response: According to reviewer’s suggestion, we have moved the citation “9” to line 62 as “In our previous study[17]”

Question 8. The sentence (lines 47-52) “In our previous study, we first hypothesized that the gut microbiota could be related to food-grinding behavior and demonstrated that the relative abundances of gut microbiome components and their metabolites are significantly correlated with food-grinding behavior in captive Brandt’s voles (Lasiopodomys brandtii) [9], a small herbivorous mammalian species that exhibits food-wasting behavior, and is found in the grasslands of Inner 51 Mongolia, China [10-13].” is really long sentence, and quite difficult to follow.

Response: We have revised this sentence as “In our previous study[17], we postulated that the gut microbiota might be associated with food grinding and demonstrated a significant correlation between the relative abundances of gut microbiome components and their metabolites with food grinding in captive Brandt's voles.” in line 62 to 65.

Question 9. Line 57 “The ultimate aim was to uncover the effect” – to ‘uncover’ or to ‘test’ (if there is any effect)?

Response: We have changed “uncover” to “test” in line 72.

Question 10. Lines 59-63 “Our findings provide guidance for reducing food waste during laboratory rodent rearing and provide new insights into the mechanisms through which the gut microbiota affect food-grinding behavior. This will ultimately broaden our understanding of the effects of vegetation biomass on food-wasting behavior in wild rodents.” It is rather part of the Discussion section, I think.

Response: According to reviewer’s suggestion, we have moved this part to part of discussion or conclusion.

Question 11. It should be used “Food grinding” phrase or “Food-grinding” phrase, in the section, and it the manuscript – not both of them (choose one of them, and use it consistently, please).

Response: According to reviewer’s suggestion, we have changed the food-grinding to food grinding throughout the manuscript.

Reviewer 3 Report

Comments and Suggestions for Authors

Han et al. investigated the effect of food restriction on food grinding by voles, as well as the potential role of the gut microbiota in this process, through a comparison of the variations between voles with different food supplies. Results indicated that food restriction can effectively inhibit food grinding. Further, Un-s-Clostridiaceae bacterium GM1, Aerococcus, and acetate content were strongly related to food grinding. However, there are many errors in citation format in the first draft, which should be revised according to the submission instructions. The introduction and discussion are too simple and need more content. Based on all of the above, the article requires minor revision before being accepted.

1.        Line 12: There is a contradiction between the unclear influencing factors and the above-mentioned influence of many factors. Please rewrite the research background.

2.        Line 23: The word “can” should be “could”.

3.        Line 40,73: The quotation is in the wrong format.

4.        The “Introduction” is too simple, so it is necessary to add background, such as the role of intestinal microorganisms in food grinding; Why choose Brandt's voles as the animal model, and why not use the varieties commonly used in the laboratory; How does the change of food supply affect the behavior of voles? The preface includes 3-5 paragraphs, and the logic is clear.

5.        In the introduction, three scientific questions are raised.

6.        Line 73: The word “represents” should be “represented”.

7.        How is 7.5 g-food supply groups designed? Is the vole used voles who completed the 15g experiment? The experimental design part is not clearly described.。

8.        Line 90,93, 113,128, etc. : Whether the citation format meets the requirements of the journal, please modify according to the standard.

9.        The content of bioinformatic analysis should be specific, for example, how are the sequences handled.

10.    Add subheading to the results section.

11.    Line 158: The word “present” should be deleted or should be “presenting”.

12.    The total number of phylum, class, order, family, genus and OTUs should be added. In addition, the composition of the top 10 or 20 fecal microorganisms in each group was supplemented at the phylum and genus levels.

13.    The discussion is thin, so you should enrich it. For example, Line 263-277 is a description of the results and has not been compared, analyzed or inferred with other studies. Adding references, for example, the first two paragraphs are only supported by two documents, which leads to monotonous content. Increase the discussion of the difference and non-difference microorganisms between the two groups, so as to infer the role in food grinding and the significance of maintaining its own stability.

14.    Line 310-311: Rewrite this sentence.

Author Response

Dear reviewer,

We are very grateful to you for your critical comments and thoughtful suggestions concerning our manuscript entitled “Effect of Food Restriction on Food Grinding in Brandt’s Voles”(ID:animals-2617770). Based on these comments and suggestions, we have made careful modifications to the original manuscript. All changes made to the text according to the comments are in red and highlighted so that they may be easily identified. We hope it has reached your magazine’s standard. All of your questions were answered below.

Question 1.    Line 12: There is a contradiction between the unclear influencing factors and the above-mentioned influence of many factors. Please rewrite the research background.

Response: We have revised this sentence to “Food grinding is supposed to be influenced by multiple-factors. However, how those factors affecting this behavior remain unclear” in line 12 to 13.

Question 2.        Line 23: The word “can” should be “could”.

Response: We have corrected “can” to “could ” in line 23.

Question 3.        Line 40,73: The quotation is in the wrong format.

Response:  We have corrected the format of these quotations in line 39 and 84

Question 4.        The “Introduction” is too simple, so it is necessary to add background, such as the role of intestinal microorganisms in food grinding; Why choose Brandt's voles as the animal model, and why not use the varieties commonly used in the laboratory; How does the change of food supply affect the behavior of voles? The preface includes 3-5 paragraphs, and the logic is clear.
Response: According to reviewer’s advice, we have added some relevant information about the influence of food supply change on behavior of voles in the section of introduction as “Food scarcity can induce alterations in rodent behavior, such as decreased activity, food hoarding, and an increase in food foraging[9]” in line 40 to 41, and some information about why chose Brandt’s vole as the study model as “Notably, wild Brandt's voles engage in wasteful behavior, gnawing on grasses without actually consuming them in the field. Remarkably, the quantity of grass wasted through non-feeding activity (excluding nesting material) accounts for a quarter of their daily food intake [16]. These feeding patterns have been shown to adversely impact the grassland ecosystems and livestock husbandry within the region [10-12]. However, the factors and mechanisms that influence this behavior have yet to be determined. Additionally, a certain percentage of captive Brandt's voles were observed strong food grinding, with the ground chow comprising up to 12% of their body mass [17], compared to mice, with orts production representing only 3~4% of their body mass [1]. As a result, we propose that captive Brandt's voles serve as a natural and viable model for researching the factors and mechanisms underlying food-wasting behavior, ultimately aiding in the conservation of grassland plants.” in line 50 to 61. Information about the role of intestinal microorganisms in food grinding is described in line 62 to 65 as “In our previous study[17], we postulated that the gut microbiota might be associated with food grinding and demonstrated a significant correlation between the relative abundances of gut microbiome components and their metabolites with food grinding in captive Brandt's voles.”. Finally, the introduction section included 3 paragraphs.

Question 5.        In the introduction, three scientific questions are raised.
Response: “The factors and mechanisms affecting food grinding are still poorly understood” is the main scientific question, which including other three scientific questions “how the food supply influences food grinding and its underlying mechanisms are not clear”, “the factors and mechanisms that influence this behavior in Brandt’s voles have yet to be determined” and “the precise role of gut microbiota in food grinding remains poorly understood”.

Question 6.        Line 73: The word “represents” should be “represented”.
Response: We have corrected the “represents” to “represented” in line 83.

Question 7.        How is 7.5 g-food supply groups designed? Is the vole used voles who completed the 15g experiment? The experimental design part is not clearly describe.
Response: According to reviewer’s advice, we have revised this part as “Of the 23 male voles, six that ground a higher average daily amount of food were marked as the high-level food-grinding voles based on the upper 27% rule [19], as well as belonging to the 15 g-food supply group. Subsequently, we conducted a food restriction experiment on this group of voles, wherein the supply of chow was reduced from 15 g to 7.5 g for a duration of 2 weeks. This new amount of chow provided was below the average daily intake (8~10 g) observed in our previous study[17]. Consequently, the voles involved in the food restriction experiment were classified as the 7.5 g-food supply group” in line 91 to 98 to make the experimental design clearer.

Question 8.        Line 90,93, 113,128, etc. : Whether the citation format meets the requirements of the journal, please modify according to the standard.
Response: We have corrected the format of these citations in line 105, 108, 134, and 149, respectively.

Question 9.        The content of bioinformatic analysis should be specific, for example, how are the sequences handled.
Response: According to reviewer’s suggestion, we have added some information about the method used to handle the sequences as “Briefly, the V4 distinct region of the prokaryotic 16S rRNA gene were amplified used specific primers (515F: 5’-GTGCCAGCMGCCGCGGTAA-3’; 806R: 5’-GGACTACHVGGGTWTCTAAT-3’). Paired-end reads were assigned to each sample referring to their unique barcodes and truncated by cutting off the barcodes and primer sequence. Paired-end reads were spliced using FLASH (V 1.2.7) to obtain the raw tags. The raw tags were filtered by QIIME (V1.9.1) and compared with the Silva database [21] using the UCHIME algorithm [22] to remove the chimera sequences, and finally obtain the effective tags.” in line 120 to 127.

Question 10.    Add subheading to the results section.
Response: According to reviewer’s suggestion, we have added the subtitles to the results section. “3.1 Differences in food intake, food ground and body mass growth rate” in line 174; 3.2 OTUs analysisin line 182; 3.3 Differences in alpha and beta diversities of fecal microbial communityin line 193;3.4 Differences in the abundances of the enriched biomarkers of fecal microbial communityin line 208-209; 3.5 The enriched KEGG pathways of fecal microbial community and differences in the content of fecal SCFAsin line 237-238;3.6 Spearman’s rank correlationsin line 270.

Question 11.    Line 158: The word “present” should be deleted or should be “presenting”.
Response: We have changed “present” to “presenting” in line 190.

Question 12.    The total number of phylum, class, order, family, genus and OTUs should be added. In addition, the composition of the top 10 or 20 fecal microorganisms in each group was supplemented at the phylum and genus levels.
Response: According to reviewer’s suggestion, we have added total number of phyla, class, order, family, genus and OTUs in line 183 to 184 as “A total of 18 phyla, 28 classes, 44 orders, 94 families, 175 genera, and 2,098 OTUs were identified in both the 15- and 7.5 g-food supply groups.”. We have added the figures of top 10 microorganisms at both the phylum and genus levels, which was named as figure 3A and 3B, respectively, and described this information in line 184 to 188 as “The dominant phyla identified were Bacteroidetes (43.97%), followed by Firmicutes (42.69%), Proteobacteria (4.66%), and Actinobacteria (3.28%) (Fig.3A). The dominant genera identified were Barnesiella(14.10%), followed by IV (7.57%), Staphylococcus (3.09%), Atopobium (2.10%), and Allobaculum (1.74%) (Fig.3B).”.

Question 13.    The discussion is thin, so you should enrich it. For example, Line 263-277 is a description of the results and has not been compared, analyzed or inferred with other studies. Adding references, for example, the first two paragraphs are only supported by two documents, which leads to monotonous content. Increase the discussion of the difference and non-difference microorganisms between the two groups, so as to infer the role in food grinding and the significance of maintaining its own stability.
Response: According to reviewer’s advices, we have added references and rewritten this part as “The abundances of certain fecal microbiota, as well as the levels of specific short-chain fatty acids (SCFAs), exhibited variation; however, there was no significant alteration observed in alpha diversity indices or the structure of the fecal microbial community following food restriction. These findings suggest that while the reduction in food supply led to a sharp decrease in food grinding behavior, it did not induce substantial changes in the gut microbiota on a large scale. Consistently, food restriction (80% the free-fed food intake) did not alter the structure of the gut microbiota in Brandt’s vole[27]. The most abundant phyla were Firmicutes, Bacteroidetes and Proteobacteria, which is consistent with the gut microbial community of Brandt’s vole studied by Xu, et al.[28]. In line with our earlier investigation comparing groups with differing degrees of food grinding, there was no significant difference in alpha diversity, with only limited dissimilarity observed in beta diversity[17]. We hypothesize that substantial alterations in gut microbiota diversity do not coincide with occurrence of food grinding. Indeed, maintaining a stable gut microbiota is highly advantageous for the host organisms[29,30].” in line 313 to 327.

Question 14.    Line 310-311: Rewrite this sentence.
Response: We have deleted this sentence because these variables were not significantly different between two groups when the method of comparison is adjusted.

Round 2

Reviewer 1 Report

Comments and Suggestions for Authors

The paper now substantially accomplished the criticisms I raised. A more extended Discussion involving rodent(s) role in ecosystem dynamics could be however advisable.

Author Response

Comment 1: The paper now substantially accomplished the criticisms I raised. A more extended Discussion involving rodent(s) role in ecosystem dynamics could be however advisable.

Response: According to reviewer’s advice, we have added one reference to make an extended discussion about the food grinding of rodents on ecosystem dynamics as “This may help to partially explain the enormous influence that rodents have on the dynamics of ecosystems by alteration of plant-herbivore interactions, which are triggered by the increase in the vegetation brought about by environmental or climatic changes[31]” in line 343 to 346.

Reviewer 2 Report

Comments and Suggestions for Authors

The manuscript has been improved and this version is significantly better than the previous one.
Still, I believe that the experimental design could be presented in a better way. In this version it is corrected; however, precise information on the group sizes (in the section 2.2. Experimental Design), could help readers to understand the design.
Additionally, still I have doubts about using so many Spearman’s rank correlations; see e.g. “P = 0.008, 0.041, 0.015, 0.050, 0.038, 0.011, 0.038, 0.035, 0.011, and 0.004” (lines 278-279 in the corrected version of the manuscript). Fortunately, in the text of the manuscript, the sample size (e.g. see legend to the Fig. 9) is unambiguously stated.

Legend to the Figure 2. The sentences “; ns means not significant” should be delated, I think.

Figure 8. It is difficult to read. It should be presented in a better way.

Author Response

Comment 1 Precise information on the group sizes (in the section 2.2. Experimental Design), could help readers to understand the design.

Response: According to reviewer’s advice, we have revised this section and added precise information on the group size as “These 6 voles made up the 15 g-food supply group” in line 101to 102.

Comment 2  Additionally, still I have doubts about using so many Spearman’s rank correlations; see e.g. “P = 0.008, 0.041, 0.015, 0.050, 0.038, 0.011, 0.038, 0.035, 0.011, and 0.004” (lines 278-279 in the corrected version of the manuscript). Fortunately, in the text of the manuscript, the sample size (e.g. see legend to the Fig. 9) is unambiguously stated.

Response: According to reviewer’s suggestion, we reduced the analysis of Spearman’s rank correlations and only analyzed the Spearman’s rank correlations between these parameters that varied significantly between groups and those not significantly varied, trying to find the relationships among them to support our conclusions. Yes, if parameters simultaneously varied significantly between groups, the correlations among them should be significantly positive or negative. Therefore, it is not necessary to do the correlation analysis among them. I am uncertain if I have fully grasped the essence of this comment. The section of Spearman’s rank correlations in the result has been revised in line 282 to 294 as “Propionatecontent was positively correlated with relative ground food, ratio of ground to consumed food, and acetate content (P = 0.033, 0.024, and 0.016, respectively; Fig. 9). Caproate content was positively correlated with relative food consumption and ground food (P = 0.012 and 0.042, respectively), acetate content (P < 0.001), and pathway enrichment of alpha-linolenic acid metabolism (P = 0.023), but negatively correlated with relative abundance of Betaproteobacteria class and Burkholderiales order (P = 0.008 and 0.011, respectively; Fig. 9). Isovalerate and valerate content were both negatively correlated with relative abundance of Betaproteobacteria class (P = 0.028 and 0.003, respectively; Fig. 9) and Burkholderiales order (P = 0.035 and 0.003, respectively). OBSP was positively correlated with relative abundance of Betaproteobacteria class and Burkholderiales order (P = 0.007 and 0.011, respectively; Fig. 9). Chao1 and ACE were both positively correlated with relative abundance of Betaproteobacteria class (P = 0.039 and 0.042, respectively; Fig. 9)”. Based on these result, the section of discussion also has been revised in line 368 to 371 as “The concentration of caproate was found to positively correlate with the relative amount of food ground, while exhibiting a decreasing trend (P = 0.065) following food restriction. This suggests that caproate may also contribute to the regulation of food grinding”, and in line 388 to 394 as “The rising trend of OBSP (P = 0.067) following food restriction should be caused by an increase in microorganisms in the Betaproteobacteria and Burkholderiales, according to the positive correlations found between OBSP and the relative abundance of these two groups of bacteria. Negative correlations between the relative abundances of Betaproteobacteria and Burkholderiales and the levels of caproate, isovalerate, and valerate suggest that these microbial taxa may inhibit the production of SCFAs within the gastrointestinal tract of Brandt's vole.”.  In section of statistical analysis, the corresponding modification is in line 177 to 179 as “Spearman's rank correlations were computed between variables that showed no significant variation and those that exhibited significant variation using R ver. 4.0.4.”.

Comment 3 Legend to the Figure 2. The sentences “; ns means not significant” should be delated, I think.

Response: We have deleted “ns means not significant” in line 236.

Comment 4. Figure 8. It is difficult to read. It should be presented in a better way.

Response: According to reviewer’s suggestion, we have modified the figure 8 and made it easy to read. The caption of figure 8 have been revised in line 276 to 277 as “Difference of fecal short-chain fatty acids (SCFAs: acetate (A), propionate (B), isobutyrate (C), butyrate (D), isovalerate (E), valerate (F), and caproate (G))”.